# Bilirubin Exerts Protective Effects on Alveolar Type II Pneumocytes in an In Vitro Model of Oxidative Stress

**DOI:** 10.3390/ijms25105323

**Published:** 2024-05-13

**Authors:** Stefanie Endesfelder, Thomas Schmitz, Christoph Bührer

**Affiliations:** Department of Neonatology, Charité—Universitätsmedizin Berlin, 13353 Berlin, Germany; thomas.schmitz@charite.de (T.S.); christoph.buehrer@charite.de (C.B.)

**Keywords:** bilirubin, hyperoxia, hypoxia, oxidative stress, alveolar epithelial cells type II

## Abstract

Newborn infants face a rapid surge of oxygen and a more protracted rise of unconjugated bilirubin after birth. Bilirubin has a strong antioxidant capacity by scavenging free radicals, but it also exerts direct toxicity. This study investigates whether cultured rat alveolar epithelial cells type II (AEC II) react differently to bilirubin under different oxygen concentrations. The toxic threshold concentration of bilirubin was narrowed down by means of a cell viability test. Subsequent analyses of bilirubin effects under 5% oxygen and 80% oxygen compared to 21% oxygen, as well as pretreatment with bilirubin after 4 h and 24 h of incubation, were performed to determine the induction of apoptosis and the gene expression of associated transcripts of cell death, proliferation, and redox-sensitive transcription factors. Oxidative stress led to an increased rate of cell death and induced transcripts of redox-sensitive signaling pathways. At a non-cytotoxic concentration of 400 nm, bilirubin attenuated oxidative stress-induced responses and possibly mediated cellular antioxidant defense by influencing Nrf2/Hif1α- and NFκB-mediated signaling pathways. In conclusion, the study demonstrates that rat AEC II cells are protected from oxidative stress-induced impairment by low-dose bilirubin.

## 1. Introduction

The fetal–neonatal transition is associated with a dramatic increase of oxygen tension in the lungs, from 20–30 mmHg (~3–5% O_2_) to about ~150 mmHg (21% O_2_, this corresponds to an alveolar oxygen partial pressure of about 100 mmHg) [1,2], which may cause considerable oxidative stress [3]. This may be more aggravated if birth occurs prematurely.

Reactive oxygen species (ROS) and the antioxidant enzyme systems, which are not yet fully developed in premature infants, are important factors in redox homeostasis [4]. ROS per se regulates various cellular signaling pathways that are essential for the maintenance of homeostasis, such as the Keap1-Nrf2-, NFκB-, and Hif1α- signaling pathways [3,5,6,7]. Oxidative stress and the resulting ROS modulate transcriptional activation and can inhibit or activate downstream signaling pathways.

In addition to the rapid rise of oxygen tension, there is a more protracted physiological increase in bilirubin concentrations in all tissues after birth. In human infants, peak bilirubin concentrations are reached at the end of the first week of life, followed by a slow decline during the ensuing month. Bilirubin is generated by the catalytic breakdown of heme, nonspecifically bound by albumin in the serum, and finally conjugated to glucuronic acid by liver cells to allow for its biliary secretion [8,9]. Hepatic glucuronidation is very slow in newborn infants, resulting in the accumulation of unconjugated bilirubin in the serum and all tissues. While virtually all newborn infants, therefore, develop some degree of jaundice, bilirubin concentrations exceeding the albumin-binding capacity are toxic and may cause transient or permanent tissue damage, notably in the brain [10,11,12,13,14,15,16]. A useful property of bilirubin can be seen in its strong antioxidant capacity as a free radical scavenger and its suitability as a cytoprotector [17,18,19], which is also supported by various in vitro and in vivo experiments [17,20]. Teleologically, the bilirubin increase may represent a physiological antioxidant to compensate for the immature redox balance in the postnatal transition period, thus closing the gap until the antioxidant system of the infant has matured [21,22,23].

In the context of oxidative stress and impairment of the developing lung, the antioxidant properties of nontoxic bilirubin levels are suggestive of a physiological protective role. However, in a study by Dani et al. [24], no correlation between the severity of neonatal respiratory distress syndrome (RDS) or bronchopulmonary dysplasia (BPD) and serum bilirubin levels was found. In an in vivo study with neonatal Gunn rats, significant antioxidant effects of bilirubin were shown [17]. However, these only existed in the circulatory compartment, where bilirubin had the highest concentration, without having protective effects on the lung tissue itself. Bilirubin levels are particularly high in the first days of life in premature infants, whereas physiological antioxidants only increase slowly [25]. Thus, bilirubin represents a component of the antioxidant capacity in the serum of newborns. Hence, bilirubin may represent a more subtle player as a serum antioxidant during the postnatal transition phase.

In this in vitro study, the effect of bilirubin on immortalized alveolar epithelial cells type II damaged by oxidative stress is investigated.

## 2. Results

### 2.1. Protective Bilirubin Concentration for Cell Viability under Oxidative Stress

The MTT test for the determination of cellular metabolic activity was used in these in vitro studies to determine the cell toxicity of applied oxygen concentrations on the AEC II and to identify the protective bilirubin concentrations in the oxidative stress-mediated cell damage model.

Hyperoxia, as the single variable, reduced the viability of AEC II, while hypoxia had no detrimental effect (0 µM bilirubin, Figure 1, left box). The MTT test showed that treatment with both higher (3 µM, 1 µM) and lower (100 nm, 50 nm, 25 nm, and 12 nm) bilirubin concentrations had no obvious effect, while treatment with 200 nm and 400 nm bilirubin led to an improvement in cell viability to control levels. As both bilirubin concentrations showed a positive effect on cell viability, a decision was made to proceed using the higher concentration (400 nm) of bilirubin in the subsequent experiments (Figure 1, right box).

### 2.2. Protective Bilirubin for Cell Survival under Oxidative Stress

Comparing the effects of low and high oxygen concentrations on the survival of AEC II compared to controls, oxidative stress induced a 2.5-fold increase in the proportion of dead cells after immunocytochemical staining after 24 h of both hypoxia (270%, *p* < 0.0001) and hyperoxia (256%, *p* < 0.0001; Figure 2). Bilirubin inhibited increased cell death to the control level of untreated cells at 21% oxygen. The complete data are presented in Appendix A.

Considering now more differentiated relevant transcripts, the transcription of *Casp3* as well as *AIF* adequately reflected this effect after one day of oxygen exposure. Hypoxia and hyperoxia increased *Casp3* expression (Figure 3a) to 138% (*p* < 0.05) and 140% (*p* < 0.05) and *AIF* expression (Figure 3b) to 176% (*p* < 0.0001) and 156% (*p* < 0.0001), respectively, compared to normoxia-exposed cells. The bilirubin used at a concentration of 400 nm reduced this increase to normoxia levels (*p* < 0.01 and 0.0001, respectively; Figure 3a,b). After a shorter exposure interval of 4 h, no effects were observed for *AIF*, but a reduced expression of *Casp3* (63%) under 80% oxygen was observed (Figure 3a). In general, *Casp3* and *AIF* expression increases over a time interval of another 20 h (Figure 3a,b). The complete data are presented in Appendix A.

The cell-cycle regulator *CycD2* was analyzed at the transcript level to test whether oxidative stress had an effect on proliferation under low or high oxygen (Figure 3c). After 4 h of incubation under hypoxia or hyperoxia, no effects of oxidative stress were observed. Bilirubin also did not affect the expression of *CycD2* after this short exposure time; 24 h after the start of the oxygen treatments, only the 80% oxygen exposure was able to reduce *CycD2* expression down to 67% (*p* < 0.05) compared to the control group (Figure 3c). Bilirubin counteracted this inhibition and led to an induction of *CycD2* expression to 183% (*p* < 0.0001). The complete data are presented in Appendix A.

### 2.3. Bilirubin Modulates Oxidative Stress-Associated Transcription Factors and Inflammatory Mediators

Oxidative stress causes corresponding cellular responses. Redox-sensitive transcription factors, such as Nrf2 or Hif1α, are an essential tool in the assessment of the redox-sensitive status of a cellular response [5,26].

After the short incubation of 4 h of the AEC II, no significant changes in *Nrf2* expression (Figure 4a) could be detected at the different oxygen concentrations, whereas after 24 h, the *Nrf2* expression was induced very impressively under 5% oxygen (153%, *p* < 0.05) as well as under 80% oxygen (182%, *p* < 0.001) compared to 21% ambient conditions. Bilirubin counteracted this effect to the control level.

*Keap1*, which sequesters Nrf2 in the cytoplasm in homeostasis, is downregulated after 4 h of incubation under hypoxia and hyperoxia (69% and 65%, respectively, Figure 4b). This effect is lost after 24 h of exposure, but bilirubin affected *Keap1* expression and increased it under 5% oxygen (261%, *p* < 0.0001) and 80% oxygen (from 69% to 128%, *p* < 0.001) compared to untreated AEC II.

GCLC, a subunit of the glutamate cysteine ligase (GCL), catalyzes the first step of glutathione biosynthesis and is transcriptionally regulated by Nrf1 and Nrf2 [27]. *GCLC* expression (Figure 4c) was induced under 80% oxygen (125%, *p* < 0.05) already after 4 h compared to normoxia-incubated cells. After a further 20 h, the expression of *GCLC* under hyperoxia only tended to be increased (139%, n.s.), whereas *GCLC* expression was now induced under 5% oxygen (150%, *p* < 0.05). Bilirubin reduced *GCLC* transcripts under 80% oxygen after 4 (80%, *p* < 0.0001) and 24 h (86%, *p* < 0.05), as well as under 5% oxygen exposure after 24 h (103%, *p* < 0.05).

The transcription factor *Hif1α* was induced under oxygen-reduced conditions at 5% both after 4 h (135%, *p* < 0.001) and persistently after 24 h (168%, *p* < 0.001) compared to AEC II exposed to 21% oxygen (Figure 4d). Hyperoxia had no effect on *Hif1α* expression. Bilirubin also exhibited inhibitory effects on *Hif1α* expression under hypoxia and reduced it to control levels. The complete data are presented in Appendix A.

One protein complex required for DNA transcription is NFκB. Oxidative stress leads to the release and nuclear translocation of NFκB, inducing the downstream transcription of pro-inflammatory mediators such as TNFα [28]. NFκB1 and NFκB2 regulate the availability of NFκB dimers, the balance of which is physiologically important [29].

NFκB subunit transcripts (Figure 5a,b) were induced under 5% oxygen (NFκB1: 192%, *p* < 0.001; NFκB2: 167%, *p* < 0.0001) and 80% oxygen (NFκB2: 135%, *p* < 0.05) after 24 h of incubation compared to normoxia. Bilirubin treatment was able to downregulate NFκB1 under hypoxia-induced (*p* < 0.001, Figure 5a) and NFκB2 (*p* < 0.05, Figure 5b) under hyperoxia-induced conditions. The NFκB1/2 transcripts were not modified at the 4 h time point.

Expression of TNFα could not be detected after 4 h, whereas induction occurred under hypoxia (191%, *p* < 0.001) and also tended to occur under hyperoxia (145%, n.s.) after a one-day exposure to oxygen (Figure 5c). The induced transcription could be inhibited by Bilirubin down to 88% (*p* < 0.001, hypoxia) and 85% (*p* < 0.05, hyperoxia).

TGFβ is activated by redox imbalances [30]. This profibinogenic cytokine (Figure 5d) was activated under hypoxic exposure after 24 h (195%, *p* < 0.001). Expression was significantly reduced by bilirubin (124%, *p* < 0.01), while a transcription inhibitory effect of TGFβ could be detected after 4 h at less than 80% (from 89% down to 57%, *p* < 0.05). The complete data are presented in Appendix A.

## 3. Discussion

The present study explores the effect of bilirubin, a potent physiological antioxidant, on hypoxia- and hyperoxia-induced injury of alveolar epithelial cell type II (AEC II). Our results indicate that cell damage with induction of associated cell-death transcripts occurred after 24 h of exposure to hypoxia or hyperoxia. This was accompanied by the activation of redox-sensitive transcription factors and mediators. Bilirubin attenuated cell damage and improved cell survival after oxidative stress injury.

The study focused on the effect of bilirubin on cultured AEC II. These cells were used because AEC IIs in the lung are responsible for surfactant production and first-line defense [31]. In newborn infants, all of which display some jaundice, malondialdehyde levels are inversely related to bilirubin concentrations [21], suggesting that higher bilirubin levels are associated with lower levels of oxidative stress. A physiological increase in bilirubin in newborns could be a transitional antioxidant after birth [21,22,23]. Hyperbilirubinemia can be harmless or harmful depending on the bilirubin level but poses a higher risk for preterm infants with very or extremely low birth weight. One of the most serious consequences of pathological hyperbilirubinemia is the development of bilirubin encephalopathy, which causes severe neurological and sensorimotor deficits [15]. Physiological neonatal hyperbilirubinemia results from low rates of glucuronidation in liver cells during the first weeks of life [9]. A possible benefit of increased bilirubin concentrations in serum could be due to the antioxidant properties of bilirubin. Bilirubin-induced neurological damage (BIND) is well described [14] and can be substantiated by any experimental studies. Little attention has been paid to studies on bilirubin effects in other cell compartments that come into contact with oxidative stress as the first barrier, such as the epithelium of the alveoli in the lungs.

The epithelium of the alveoli consists largely (95%) of alveolar cells type I (AEC I), which are responsible for gas exchange from the alveoli to the capillaries of the vascular system. A small proportion of pulmonary epithelial cells are type II cells, which are responsible for surfactant production and can replace damaged type I cells by differentiation [31]. In general, AEC IIs are less susceptible to damage than AEC Is, but increased oxygen concentrations and oxidative stress can still impair their function. They secrete inflammation and oxidative stress-response modifying cytokines and proteins.

The main result of this in vitro study is that bilirubin inhibits redox-sensitive transcription factors in the reverse direction to activation under oxidative stress. When oxidative stress acts, free radicals (ROS) are formed in toxic concentrations, which can then either damage macromolecules or activate signaling cascades that induce a regular antioxidant stress response. Both modulate and regulate essential cellular processes, such as proliferation, cell death, and differentiation. The pathophysiology of lung diseases is due to an imbalance in the reduction–oxidation equilibrium. The therapeutic potential regarding influencing redox homeostasis has great potential for the postnatal treatment of respiratory diseases. Dynamic fluctuations in alveolar pO_2_ and the associated changes in redox status influence pulmonary gene expression. The main hypoxia-inducible transcription factors Hif1α, Nrf2 as the first-line transcription factor of defense against oxidative stress, and NFκB as a downstream redox-sensitive transcription factor are particularly important. They coordinate the adaptive homeostatic response to oxidative stress. A corresponding induction of redox-sensitive transcription factors was shown for Nrf2 in hyperoxia and for Hif1α in hypoxia, as well as for NFκB for both hyperoxic and hypoxic exposure in our in vitro study. Hif1α physiologically coordinates the adaptive homeostatic response under low oxygen partial pressures, and in particular, the alveolar epithelium responds [32]. The Nrf2-Keap1 system plays an important role in hyperoxia. Exposure to high oxygen concentrations induced an increase in Nrf2 mRNA levels [33] and DNA binding activity [34] in rodents. Downstream target genes also increased transcription [34]. In Nrf2^−/−^ mice, toxic oxygen-induced lung injury was enhanced, whereas an increase in Nrf2 expression had a protective effect [35]. The interaction between NFκB and ROS appears to be more complex. Oxidative changes can lead to the activation of NFκB and, thus, induce the transcription of both antioxidative and prooxidative genes. The role of NFκB is double-edged, as it activates hyperoxia-protective and -progressive-enhancing target genes [7,36,37]. Consistent with this, the expression of the oxidative stress marker GCLC, which is transcriptionally regulated by Nrf2, increased after a short incubation under high oxygen concentrations [27]. To restore the redox balance in the face of an oxidative challenge, an adaptive equilibrium requires a cross-connection between signaling pathways. The glutathione signaling pathway is critical for providing an equilibrium interface between oxidative stress and adaptive responses of cell protection [38]. However, prolonged incubation of AEC II under oxygen deprivation also led to the induction of GCLC. The first step of short-term glutathione synthesis consists of the formation of γ-glutamylcysteine (γ-GC), catalyzed by a heterodimer of GCL consisting of the catalytic (GCLC) and modifying subunit (GCLM) [39]. The GCLM mRNA has a Hif1a binding site that is responsible for GCLM induction and, thus, the regulation of GCL [40].

Oxidative stress and the triggering of cell death are well known, and in the alveolar epithelial cells, both oxygen deficiency and oxygen oversupply lead to an increased cell death rate. The 24 h hypoxia, and also the hyperoxia, caused a change in cell-death-associated transcripts. An increase in Casp3 gene expression favors caspase-dependent apoptosis, and the increased transcripts for AIF also involve caspase-independent cell-death signaling pathways. Changes in redox status in certain organelles, such as the mitochondria, modulate apoptosis-associated mediators. For example, oxygenated cytochrome c initiates the intrinsic apoptosis signaling pathway [41]. Apoptosis of alveolar epithelial cells also characterizes hyperoxia-induced acute lung injury in preterm infants, as shown in experimental studies [42,43]. Interestingly, cyclinD2 gene expression was explicitly inhibited by 80% exposure. The maintenance of the cell cycle is tightly controlled by interactions between cyclins and cyclin-dependent kinases, such as cyclinD1 or cyclinD2. Redox-sensitive cysteine residues are sensitive to changes in intracellular redox homeostasis [44]. The activation of NFκB is a response to various stimuli, such as cytokines or ROSs, and is organized sequentially at the molecular level, whereby there is also a correlation between NFκB activation and activation of the expression of redox-sensitive enzymes [45,46]. TNFα as a proinflammatory cytokine modulates the inflammatory response. TGFβ is crucial for the regulation of tissue homeostasis under physiological conditions, but under pathological conditions, it plays a crucial role in regulating the progression of inflammation [47]. Both cytokines are induced by hyperoxic insult, while TNFα is only activated under high oxygen in immortalized AEC II.

Unconjugated bilirubin can diffuse into any cell, where it may exert neurotoxic [14] or antioxidant [17,25,48] effects in a concentration-dependent manner. Here, we showed that bilirubin protectively modulated oxidative stress responses in an alveolar epithelial cell line in both hypoxia- and hyperoxia-induced models. When glutathione, the most important endogenous intracellular antioxidant, or bilirubin is deprived of HeLa cells, bilirubin proves to be a relevant cytoprotector [49]. Liu et al. showed that treatment with biliverdin reductase reduced the clinical and pathological signs of experimental autoimmune encephalomyelitis than other antioxidants [48]. It is unclear whether these antioxidant effects are mediated exclusively via free-radical-scavenging properties or via a network of different signaling pathways. The literature data for effects on lung tissue or pulmonary cells for bilirubin are scarce. When comparing different cell types, each cell type has different thresholds for bilirubin concentration that are responsible for a protective or toxic effect. In the present study, we approached this concentration by determining cell viability and were able to define a concentration of bilirubin of 400 nM for immortalized AEC II. For hippocampal neuronal cells, protective bilirubin concentrations were between 10–50 nM [50]. In a human endothelial cell line, the intracellular antioxidant activity of bilirubin was shown to be about 11 nM, which corresponds to about 13 nM of free serum bilirubin [51]. In this initial study, we tested a bilirubin concentration at which cell viability was preserved in the oxidative stress model. It cannot be excluded that a lower bilirubin concentration could also have antioxidant effects on AEC II. However, the exact concentration ranges between the antioxidative and pro-oxidative effects of bilirubin remain undetermined.

In summary, alveolar epithelial cells are vulnerable to oxygen oscillations. The attempt to balance and restore a physiologically necessary antioxidant–oxidative balance activates redox-sensitive signaling pathways. Under non-physiological conditions, i.e., when the immature lung comes into contact with atmospheric oxygen too early, an interplay of these modulators begins. Bilirubin attenuated oxidative stress-induced responses and possibly mediated cellular antioxidant defenses by influencing the Nrf2/Hif1α and NFκB-mediated signaling pathways. AEC IIs are essential in the repair and regeneration of the alveolar epithelium. Reducing oxidative-induced damage to AEC II preserves the differentiation capacity to AEC I. The in vitro study on AEC II is expected to highlight the significance of bilirubin as a key component of the physiological antioxidant capacity in the sera of newborn and premature infants, especially when compared to pulmonary influences, and may also open up the possibility of further investigations into bilirubin as a serum antioxidant during the postnatal transition phase of the immature lung.

## 4. Materials and Methods

### 4.1. Cell Culture: Cell Line AEC II/ RLE-6TN

Rat type II alveolar cells (AEC II/ RLE-6TN) obtained from the America Tissue Type Collection (ATTC, Manassas, VA, USA, cat. CRL-2300) were cultured using Ham’s F12 medium (Biochrom GmbH, Berlin, Germany, cat. FG0815) containing 2 mM L-glutamine and supplemented with 10% Fetal Bovine Serum (PanBiotech GmbH, Aidenbach, Germany, cat. 0522D), 0.01 mg/mL bovine pituitary extract (Gibco™, Thermo Fisher Scientific, Waltham, MA, USA, cat. 13028-014), 0.005 mg/mL insulin (Sigma-Aldrich, St. Louis, MO, USA, cat. I-6634), 2.5 ng/mL insulin-like growth factor (Sigma-Aldrich, cat. I-3769), 0.00125 mg/mL transferrin (Sigma-Aldrich, cat. I-1147), and 2.5 ng/mL epidermal growth factor (Sigma-Aldrich, cat. I-9644). Cells were grown at 37 °C in a 5% CO_2_ humidified atmosphere. The medium was changed every 2–3 days and the cells were subcultured when at 80–90% confluence. AEC II were sown in complete Ham’s F12 medium at a density of 2.5 × 10^4^ cells/cm^2^ and cultured for 24 h prior to treatment.

### 4.2. Determining the Oxygen Level and Relevance of the Cellular Model

The cell-culture studies carried out on AEC IIs were performed under three different oxygen concentrations. These were colloquially labeled with the common terms “hypoxia” (5% oxygen), “normoxia” (21% oxygen), and “hyperoxia” (80% oxygen). However, the actual oxygen tension perceived by the cells depends on the cell density, the level of the medium supernatant, and the incubation time. Nevertheless, the applied oxygen tensions are in the range of physiological physioxia (3–8% oxygen) and normoxia (18% oxygen). Hyperoxic ambient oxygen induces an approximately fourfold increase in oxygen concentration and induces oxidative stress [52].

### 4.3. Oxygen Exposure and Bilirubin Administration

The cells were exposed to different oxygen concentrations (5%, 21%, and 80%) in a suitable cell-culture incubator with an oxygen sensor (Binder GmbH, Tuttlingen, Germany) one day after sowing (DIV 1), depending on the experimental assignment. Two exposure times were set at 4 h and 24 h, while the cells were left untreated or treated with bilirubin (BR) in the medium. A schematic representation of the experimental design is displayed in Figure 6.

For the initial experiments, an MTT cell viability assay (see Section 4.6) was used to identify the optimal dose of bilirubin (see Section 2.1), i.e., the selection was based on a bilirubin concentration that counteracts the oxidative stress-induced loss of viability. All further analyses are carried out under the three different oxygen concentrations and the selected bilirubin concentration in the medium.

Bilirubin/BR (Sigma-Aldrich, cat. B4126) was dissolved in 0.1 N NaOH and sterilely filtered. A bilirubin stock solution was then adjusted to a concentration of 1 mM with sterile water. Aliquots of this stock solution could be stored at −20 °C in dark tubes until further use. On the day of the experiments, a fresh bilirubin working solution was prepared with bovine serum albumin (BSA, Carl Roth, Karlsruhe, Germany, cat. 8076), which is dissolved at 3% (*w*/*v*) in phosphate-buffered saline (PBS, pH 7.4) and sterilely filtered. Further dilutions to adjust the final bilirubin concentrations were carried out in a complete cell-culture medium. To avoid bilirubin photoisomerization, all studies were conducted in low light.

### 4.4. RNA-Extraction and qPCR

As previously described [33,42], total RNA was isolated by acidic phenol–chloroform extraction using an RNA Solv Reagent (Omega Bio-Tek, Norcross, GA, USA, cat. R6830) following the manufacturer’s instructions. Total RNA (2 µg) was reverse transcribed, and DNase treated. The PCR products of genes of interest were quantified in real time using qPCRBIO Probe Mix Lo-ROX (PCR Biosystems, London, UK, cat. PB20.21) and dye-labeled fluorogenic reporter oligonucleotide probes of the sequences listed in the table (Table 1). The PCR amplification was performed in 96-well optical reaction plates subjected to 40 cycles of 5 s at 95 °C and 25 s at 60 °C each. The QuantStudio™ 3 Real-Time PCR System (Applied Biosystems by Thermo Fisher Scientific Inc., Waltham, MA, USA) was used to quantify the expression of the target genes. The analysis was performed using the 2^−ΔΔCT^ method [53], with hypoxanthine phosphoribosyltransferase 1 (HPRT) serving as an internal reference.

### 4.5. EarlyTox Live/Dead Assay

The detection with the EarlyTox Live/Dead Assay Kit (Molecular Devices, Sunnyvale, CA, USA, cat. R8341) is based on two markers that are used to detect living and dead cells. Calcein AM is the marker for living cells, while EthD-III is the marker for dead cells. Calcein AM is not fluorescent per se. If calcein AM penetrates the intact cell membrane, it is converted into the fluorescent form of calcein by intracellular esterases and living cells will fluorescent green in the cytosol. EthD-III is also non-fluorescent but, in contrast to calcein AM, is impermeable to an intact plasma membrane. When cells die, they lose their cell-membrane integrity and EthD-III can penetrate the cell. Now EthD-III can bind to nucleic acids and the dead cells fluoresce bright red.

The AEC IIs were cultivated according to the experimental design on coverslips (12 mm, neoLab Migge GmbH, Heidelberg, Germany, cat. WQ-0657) and placed in 24-well Falcon^®^ plates (Corning GmbH, Kaiserslautern, Germany, cat. 353047). The analyses of the experiments were carried out according to the manufacturer’s instructions and 24 h after the start of exposure. The coverslips with the stained cells were embedded on slides using PBS and immediately viewed blinded using a Keyence BZ 9000 compact-fluorescence microscope with BZ-II Viewer software 1.1.2.4 (Keyence, Osaka, Japan). The cells were counted manually by using Adobe Photoshop software 22.0.0 (Adobe Systems Software Ireland Limited, Dublin, Ireland). The analyses were carried out using the counted cells, which were counted from four non-overlapping image sections, whereby the proportion of dead cells was shown in relation to the living cells.

### 4.6. MTT

The 3-(4,5-Dimethylthiazol-2-yl)-2,5-diphenyltetrazolium bromide-Assay (MTT; Sigma-Aldrich, cat. 475989) was used to evaluate cell viability [54]. For this purpose, AEC IIs were seeded in 96-well plates and incubated for 24 h at 37 °C and 5% CO_2_ in a complete medium. On the day of the experiment, the medium was completely removed and replaced by a medium with or without bilirubin. According to the experimental design, the cells were incubated for 24 h at 5%, 21%, or 80% oxygen. Then, the medium was removed, the cells were washed once with PBS, and a fresh medium containing MTT solution (final concentration of 0.5 mg/mL) was added. After 4 h, to solubilize the formazan, a combination of detergent and organic solvent (dimethyl sulfoxide, Sigma-Aldrich, cat. D8418; sodium dodecyl sulfate, Sigma-Aldrich, cat. 436143; glacial acetic acid, Sigma-Aldrich, cat. 320099) was added and shaken for 15 min. The absorbance of the AEC II was determined at 570 nm using a microplate reader. The AEC II under normoxia without bilirubin was set as 100%, and all other measurements were expressed as a percentage relative to the value of the control cells.

### 4.7. Statistical Analyses

All experiments were carried out simultaneously in comparable cell-culture passages, whereby 6 independent experiments were always carried out in succession (n = 6/experimental group). As previously described [55], all box and whisker plots represent the interquartile range (box), with the line representing the median, while whiskers show the data variability outside the upper and lower quartiles. This is except for the MTT test, which is present as the mean ± standard error of the mean and 5 independent experiments. The control and experimental groups were compared using a one-way analysis of variance (ANOVA). All data sets were tested for non-Gaussian distribution or non-equal variances, with the result that all data sets could be analyzed using a one-way ANOVA. In addition, the Grubbs outlier test was used to detect outliers from normal distributions, and multiple mean comparisons were performed using the Bonferroni post hoc test. A *p* value of <0.05 was considered significant. All graphics and statistical analyses were performed using the GraphPad Prism software 8.4.3 (GraphPad Software, La Jolla, CA, USA).

## Figures and Tables

**Figure 1 ijms-25-05323-f001:**
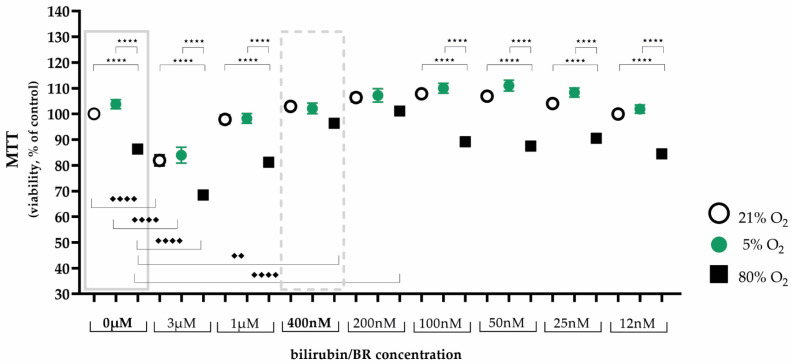
Illustration of the effects of hypoxia and hyperoxia on the cell viability of AEC II without and with different bilirubin/BR concentrations after 24 h of incubation. Quantitative analysis was conducted for the untreated and bilirubin-treated cells with concentrations from 3 µM to 12 nm under exposure of normoxia with 21% oxygen (100% control, white circle), hypoxia with 5% oxygen (green circle), and hyperoxia with 80% oxygen (black square). Data are normalized to the level of cells exposed to normoxia (21% O_2_) without bilirubin in the medium. n = 5 individual experiments/group. ^★★★★^ *p* < 0.0001 vs. oxygen concentrations with identical bilirubin concentration; ♦♦ *p* < 0.01, ♦♦♦♦ *p* < 0.0001 identical oxygen concentrations with different bilirubin concentrations (ANOVA, Bonferroni’s post hoc test).

**Figure 2 ijms-25-05323-f002:**
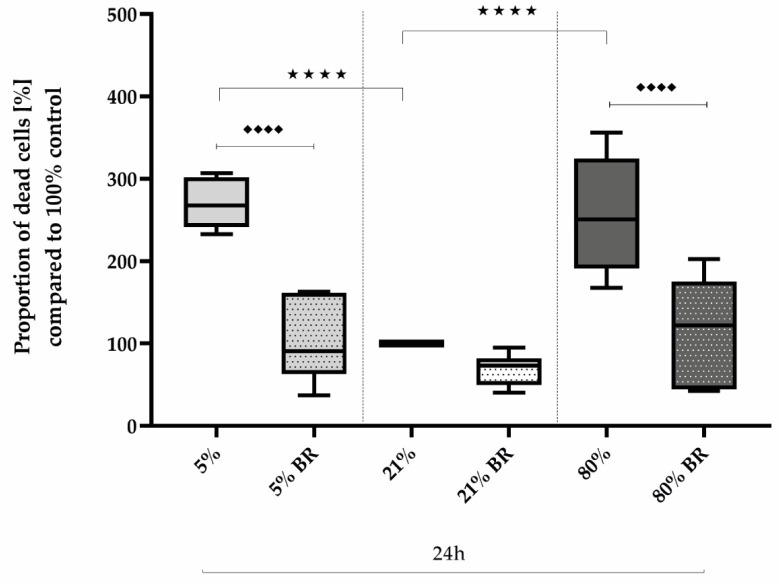
Quantitative analysis of the effects of hypoxia (light-gray boxes) and hyperoxia (dark-gray boxes) on the cell death of AEC II without (no hatching) and with bilirubin/BR (hatching) compared to normoxia control (white boxes). Data are normalized to the level of cells exposed to normoxia (21% O_2_) without BR in the medium. n = 6 individual experiments/group. ^★★★★^ *p* < 0.0001 vs. different oxygen concentrations without BR; ♦♦♦♦ *p* < 0.0001 vs. identical oxygen concentrations with BR (ANOVA, Bonferroni’s post hoc test).

**Figure 3 ijms-25-05323-f003:**
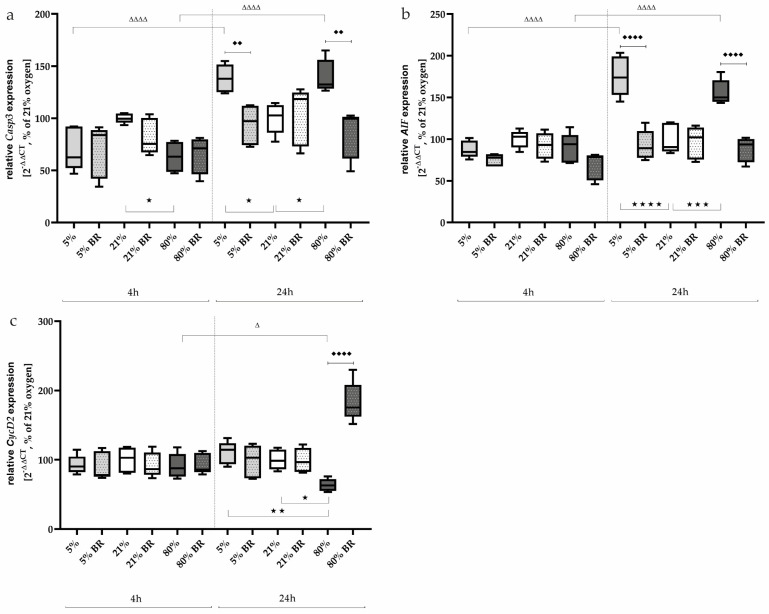
Quantitative analysis of the effects of hypoxia (light-gray boxes) and hyperoxia (dark-gray boxes) on the RNA expression of AEC II without (no hatching) and with bilirubin/BR (hatching) compared to normoxia control (white boxes) for (**a**) Casp3, (**b**) AIF, and (**c**) CycD2. Data are normalized to the level of cells exposed to normoxia (21% O_2_) without BR in the medium. n = 6 individual experiments/group. ^★^ *p* < 0.05, ^★★^ *p* < 0.01, ^★★★^ *p* < 0.001, ^★★★★^ *p* < 0.0001 vs. different oxygen concentrations without BR within one exposure time; ♦♦ *p* < 0.01, ♦♦♦♦ *p* < 0.0001 vs. identical oxygen concentrations with BR; ^Δ^ *p* < 0.05, ^ΔΔΔΔ^ *p* < 0.0001 vs. identical oxygen concentrations without BR outside one exposure time (ANOVA, Bonferroni’s post hoc test).

**Figure 4 ijms-25-05323-f004:**
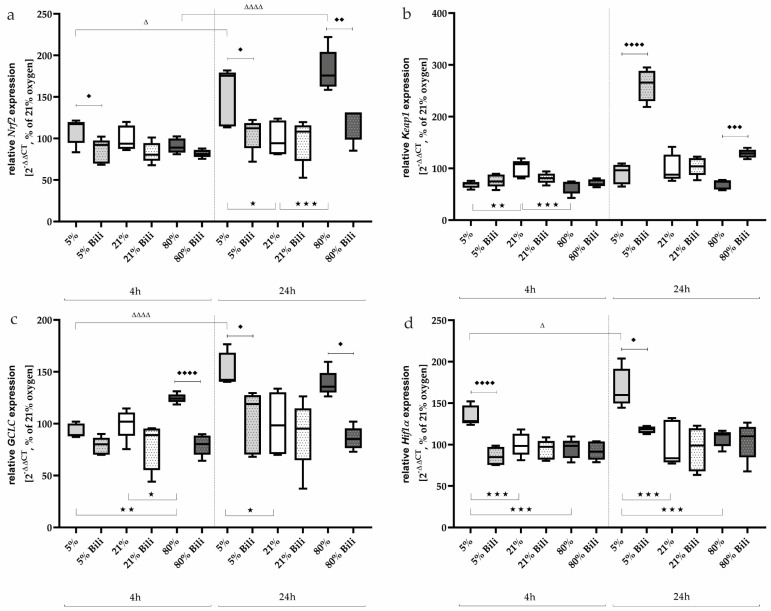
Quantitative analysis of the effects of hypoxia (light-gray boxes) and hyperoxia (dark-gray boxes) on the RNA expression of AEC II without (no hatching) and with bilirubin/BR (hatching) compared to normoxia control (white boxes) for (**a**) Nrf2, (**b**) Keap1, (**c**) GCLC, and (**d**) Hif1α. Data are normalized to the level of cells exposed to normoxia (21% O_2_) without BR in the medium. n = 6 individual experiments/group. ^★^ *p* < 0.05, ^★★^ *p* < 0.01, ^★★★^ *p* < 0.001 vs. different oxygen concentrations without BR within one exposure time; ♦ *p* < 0.05, ♦♦ *p* < 0.01, ♦♦♦ *p* < 0.001, ♦♦♦♦ *p* < 0.0001 vs. identical oxygen concentrations with BR; ^Δ^ *p* < 0.05, ^ΔΔΔΔ^ *p* < 0.0001 vs. identical oxygen concentrations without BR outside one exposure time (ANOVA, Bonferroni’s post hoc test).

**Figure 5 ijms-25-05323-f005:**
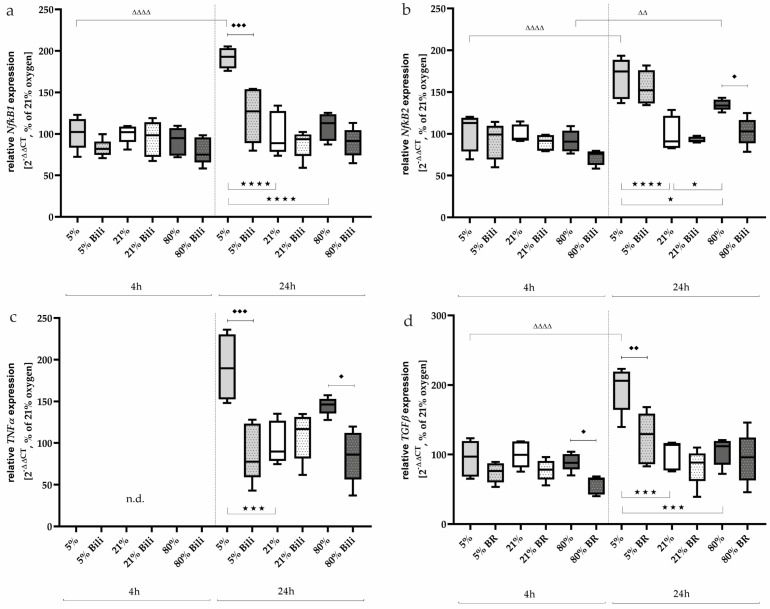
Quantitative analysis of the effects of hypoxia (light-gray boxes) and hyperoxia (dark-gray boxes) on the RNA expression of AEC II without (no hatching) and with bilirubin/BR (hatching) compared to normoxia control (white boxes) for (**a**) NfκB1, (**b**) NfκB2, (**c**) TNFα, and (**d**) TGFβ. Data are normalized to the level of cells exposed to normoxia (21% O_2_) without BR in the medium. n = 6 individual experiments/group. ^★^ *p* < 0.05, ^★★★^ *p* < 0.001, ^★★★★^ *p* < 0.0001 vs. different oxygen concentrations without BR within one exposure time; ♦ *p* < 0.05, ♦♦ *p* < 0.01, ♦♦♦ *p* < 0.001 vs. identical oxygen concentrations with BR; ^ΔΔ^ *p* < 0.01, ^ΔΔΔΔ^ *p* < 0.0001 vs. identical oxygen concentrations without BR outside one exposure time (ANOVA, Bonferroni’s post hoc test).

**Figure 6 ijms-25-05323-f006:**
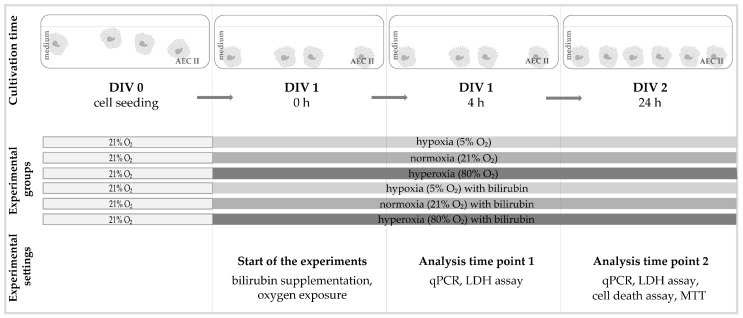
Experimental setup for the cell culture of this study. The AEC IIs were seeded in the appropriate number of cells (DIV 0). After 24 h under normal culture conditions (DIV 1), the medium was completely changed and added according to the experimental groups with or without bilirubin and exposed to the corresponding oxygen conditions. The cell cultures were incubated under the experimental conditions for up to 4 h (DIV 1, 4 h) or 24 h (DIV 2, 24 h) without any further changes and then analyzed immediately afterward.

**Table 1 ijms-25-05323-t001:** Sequences of oligonucleotides.

	Oligonucleotide Sequence 5′–3′	Accession No.
	AIF	
forward	CACAAAGACACTGCAGTTCAGACA	NM_031356.1
reverse	AGGTCCTGAGCAGAGACATAGAAAG	
probe	6–FAM–AGAAGCATCTATTTCCAGCC–TAMRA	
	Casp3	
forward	ACAGTGGAACTGACGATGATATGG	NM_012922.2
reverse	AATAGTAACCGGGTGCGGTAGA	
probe	6–FAM–ATGCCAGAAGATACCAGTGG–TAMRA	
	CycD2	
forward	CGTACATGCGCAGGATGGT	NM_199501.1
reverse	AATTCATGGCCAGAGGAAAGAC	
probe	6–FAM–TGGATGCTAGAGGTCTGTGA–TAMRA	
	GCLC	
forward	GGAGGACAACATGAGGAAACG	NM_012815.2
reverse	GCTCTGGCAGTGTGAATCCA	
probe	6–FAM–TCAGGCTCTTTGCACGATAA–TAMRA	
	Hif1α	
forward	GCGCCTCTTCGACAAGCTT	NM_024359.2
reverse	CTGCCGAAGTCCAGTGATATGA	
probe	6–FAM–AGAGCCCGATGCCCTGACTCTGCT–TAMRA	
	HPRT	
forward	GGAAAGAACGTCTTGATTGTTGAA	NM_012583.2
reverse	CCAACACTTCGAGAGGTCCTTTT	
probe	6–FAM–CTTTCCTTGGTCAAGCAGTACAGCCCC–TAMRA	
	Keap1	
forward	GATCGGCTGCACGGAACT	NM_057152.2
reverse	GCAGTGTGACAGGTTGAAGAACTC	
probe	6–FAM–CTCGGGAGTATATCTACATGC–TAMRA	
	MCP1	
forward	AGCATCCACGTGCTGTCTCA	NM_031530.1
reverse	GCCGACTCATTGGGATCATC	
probe	6–FAM–AGATGCAGTTAATGCCCCAC–TAMRA	
	MIP2	
forward	CCTACCAAGGGTTGACTTCAAGA	NM_053647.1
reverse	GCTTCAGGGTTGAGACAAACTTC	
probe	6–FAM–AGACAGAAGTCATAGCCACT–TAMRA	
	NfκB1	
forward	GACCCAAGGACATGGTGGTT	NM_001276711.1
reverse	TCATCCGTGCTTCCAGTGTTT	
probe	6–FAM–CTGGGAATACTTCACGTGAC–TAMRA	
	NfκB2	
forward	GCCTAAACAGCGAGGCTTCA	NM_001008349.1
reverse	TCTTCCGGCCCTTCTCACT	
probe	6–FAM–TTTCGATATGGCTGTGAAGG–TAMRA	
	Nrf2	
forward	ACTCCCAGGTTGCCCACAT	NM_031789.2
reverse	GCGACTCATGGTCATCTACAAATG	
probe	6–FAM–CTTTGAAGACTGTATGCAGC–TAMRA	
	TGFβ	
forward	CCTGCAGAGATTCAAGTCAACTGT	NM_021578.2
reverse	GTCAGCAGCCGGTTACCAA	
probe	6–FAM–CAACAATTCCTGGCGTT–TAMRA	
	TNFα	
forward	CCCCCAATCTGTGTCCTTCTAAC	NM_012675.2
reverse	CGTCTCGTGTGTTTCTGAGCAT	
probe	6–FAM–TAGAAAGGGAATTGTGGCTC–TAMRA	

Abbreviations: apoptosis-inducing factor (AIF), caspase−3 (Casp3), cyclin D2 (CycD2), glutamate−c ligase catalytic subunit (GCLC), hypoxia-inducible factor 1α (Hif1α), hypoxanthine phosphoribosyltransferase 1 (HPRT), Kelch-like ECH-associated protein 1 (Keap1), C-C motif chemokine ligand 2 (MCP-1), C-X-C motif chemokine ligand 2 (MIP-2), nuclear factor kappa B subunit 1/2 (NFκB1/2), nuclear factor erythroid 2−related factor 2 (Nrf2), transforming growth factor β (TGFβ), and tumor necrosis factor (TNFα), 6−carboxyfluorescein (6−FAM), and tetramethylrhodamine (TAMRA).

## Data Availability

The data used to support the findings of this study are available from the corresponding author upon request. The analyzed data used to create the graphs and statistical evaluation are attached in the Appendix A of this work.

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
