# Peer review of "Bilirubin Exerts Protective Effects on Alveolar Type II Pneumocytes in an In Vitro Model of Oxidative Stress"

_ijms, 2024, doi:10.3390/ijms25105323_

Round 1
Reviewer 1 Report
Comments and Suggestions for Authors
Endesfelder et al. wrote an article about a vitally important issue regarding newborns. The article is comprehensive, well-written, the topic is particularly relevant today.
However, some questions came to my mind after reading:
1. In the abstract, the exact doses of effective bilirubin is missing in the conclusion.
2. in line 55. what does BPD mean? Please write the abbreviation.
3. Figures are well edited, transparent; however, in my opinion an additional figure of the experimental protocol could significantly improve the quality of the paper.
4. What other cell types could be used in this experiment that would be of similar importance in neonatal pathology?
5. MDA is also a great indicator of oxidative stress besides ROS. It would have been preferable to measure as well.
Reviewer 2 Report
Comments and Suggestions for Authors
The manuscript:“Bilirubin exerts protective effects on alveolar type II pneumocytes in an in vitro model of oxidative stress” describes the effect of bilirubin on immortalized alveolar epithelial cells type II damaged by oxidative stress in vitro study. The manuscript is well organized and it represent a good contribution to the field of interest for oxidative stress.
It is evident from the reported literature that the authors are experts in oxidoreductive mechanisms in different biological models.
The in vivo association between bilirubin and chronic respiratory diseases is also evident from the literature. (doi:10.2147/COPD.S360485)
The specific question related to the manuscript is the following:
Can the authors verify the method of oxidative stress induced with different percentages of oxygen with another method for example by examining the effects of a classical reactive oxygen species H2O2 to verify the method?
If this question is excluded, I recommend that the manuscript can be accepted.
